# Preparation of optimized concanavalin A-conjugated Dynabeads® magnetic beads for CUT&Tag

Yasuhiro Fujiwara[1]*, Yuji Tanno[2], Hiroki Sugishita[3], Yusuke Kishi[3], Yoshinori Makino[1], Yuki Okada[1]

**1** Institute of Quantitative Biosciences, The University of Tokyo, Tokyo, Japan, **2** Bioscience Department, VERITAS Corporation, Tokyo, Japan, **3** Graduate School of Pharmaceutical Sciences, The University of Tokyo, Tokyo, Japan

* y.fujiwara@iqb.u-tokyo.ac.jp

**Data Availability Statement:** All NGS sequence files are available from the NCBI GEO database (GSE182254).

## Abstract

Epigenome research has employed various methods to identify the genomic location of proteins of interest, such as transcription factors and histone modifications. A recently established method called CUT&Tag uses a Protein-A Tn5 transposase fusion protein, which cuts the genome and inserts adapter sequences nearby the target protein. Throughout most of the CUT&Tag procedure, cells are held on concanavalin A (con A)-conjugated magnetic beads. Proper holding of cells would be decisive for the accessibility of Tn5 to the chromatin, and efficacy of the procedure of washing cells. However, BioMag®Plus ConA magnetic beads, used in the original CUT&Tag protocol, often exhibit poor suspendability and severe aggregation. Here, we compared the BioMag beads and Dynabeads® magnetic particles of which conjugation of con A was done by our hands, and examined the performance of these magnetic beads in CUT&Tag. Among tested, one of the Dynabeads, MyOne-T1, kept excessive suspendability in a buffer even after overnight incubation. Furthermore, the MyOne-T1 beads notably improved the sensitivity in CUT&Tag assay for H3K4me3. In conclusion, the arrangement and the selection of MyOne-T1 refine the suspendability of beads, which improves the association of chromatin with Tn5, which enhances the sensitivity in CUT&Tag assay.

## Introduction

Over the past decade, understanding of transcriptional regulation through regulatory factors, including transcription factors (TFs) and histone modifications, has been accelerated along with the invention and spread of chromatin immunoprecipitation with sequencing (ChIP-seq) method. ChIP-seq has been a primary method to detect genomic location of proteins of interest in biological research. However, several limitations have been indicated; requiring a large number of cells and costly deep sequencing, owing to solubilization of the entire genome of cells, which indeed generate high background reads. To overcome these limitations, Dr. Henikoff's group invented Cleavage Under Targets and Release Using Nuclease (CUT&RUN),

**Funding:** This work was supported in part by JSPS KAKENHI grant number 17K15392 and 20H00446 (to Y.F.) and JST ERATO (JPMJER1901 to Y.O.). Y. T. is employed by VERITAS corporation. The funder provided support in the form of salaries for Y.T., but did not have any additional role in the study design, data collection and analysis, decision to publish, or preparation of the manuscript. The specific roles of Y.T. are articulated in the 'author contributions' section.

**Competing interests:** The authors report that there were material support and technical advice from VERITAS corporation for this study that could have influenced its outcome and that one of the authors, Y.T. is its employee. This does not alter our adherence to PLOS ONE policies on sharing data and materials." We believe our revised manuscript would suffice the reviewer's concerns. It would be greatly appreciated if the editorial office and reviewers could consider this revision. All the authors have read the manuscript and have approved this submission. All study participants provided informed consent, and the study design was approved by an ethics review board.

which used antibodies and a Protein A-Micrococcal Nuclease (pA/MNase) fusion protein on unfixed cells [1,2]. The same group further introduced a derived Cleavage Under Targets and Tagmentation (CUT&Tag), which utilized a Protein A-Tn5 transposase (pA/Tn5) fusion protein instead of PA/MNase [3,4]. CUT&RUN and CUT&Tag require a remarkably reduced number of cells, while the sequence profiling resolution is remarkably increased. In both methods, cells are bound to concanavalin A (con A)-coated magnetic beads and are handled throughout the process until the DNA extraction step. To date, options for commercially available con A-coated magnetic beads were limited at this point. Further, the con A-coated magnetic beads, BioMag®Plus ConA magnetic beads (hereafter BioMag), used in the original protocols, are difficult to handle because of their poor suspendability and severe aggregation. One might argue that the efficiency of enzyme reaction in either protocol would be suffered.

To solve this issue, we evaluated different kinds of Dynabeads® magnetic beads (hereafter Dynabeads) that vary in size, cell binding capacity, or water binding capacity in order to verify whether the Dynabeads can be an alternative choice for the conventional con A-coated magnetic beads.

## Materials and methods

### Ethics declarations

Animal experiments in this study were approved by the Animal Experiment Ethics Committees at Institute for Quantitative Biosciences, University of Tokyo (approval #2715, #2809). Experiments were performed in precise accordance with the manual provided by the Life Science Research Ethics and Safety Committee, the University of Tokyo.

### Animal experiments and ethics statement

C57BL/6J mice (CLEA Japan, Tokyo, Japan) were used for animal experiments. No statistical method was used to estimate the sample size.

### Preparation of con A-coated beads

Four different streptavidin-conjugated Dynabeads, M-270, M-280, MyOne C1, and MyOne T1 that are capable of binding to biotin-conjugated concanavalin A (con A) were purchased from Thermo Fisher (**Table 1**). To conjugate con A, 100 μL of each beads is washed with 1× PBS (pH 6.8) for three times and resuspended in 100 μL of 1× PBS (pH 6.8) containing 0.01% Tween-20. Fifty μL of biotin-conjugated con A solution (2.3 mg/mL, Sigma Aldrich, C2272) in 1× PBS (pH 6.8) containing 0.01% Tween-20 is added to the beads, and rotated for 30 min at room temperature (RT). The beads are briefly spun down, and the supernatant is removed and used for measuring unbound con A in a "con A binding assay." The remaining beads are resuspended in 100 μL of 1x PBS (pH 6.8) containing 0.01% Tween-20, then used for "cell-binding

**Table 1. Comparison of Streptavidin conjugated magnetic beads.**

| Beads | Binding capacity to free biotin | Diameter of beads | Water binding capacity |
|---|---|---|---|
| Dynabeads M-280 Streptavidin | 650–900 pmol/mg beads | 2.8 μm | Hydrophobic |
| Dynabeads M-270 Streptavidin | 650–1350 pmol/mg beads | 2.8 μm | Hydrophilic |
| Dynabeads MyOne Streptavidin C1 | >2500 pmol/mg beads | 1.05 μm | Hydrophilic |
| Dynabeads MyOne Streptavidin T1 | >1300 pmol/mg beads | 1.05 μm | Hydrophobic |
| BioMag Plus Concanavalin A | NA | 1.0 μm | not known |

assay" or the CUT&Tag procedure. BioMag from the original CUT&Tag protocol (Bangs Laboratories, BP531) [3,4] is used for comparison.

## Con A binding assay

Nanodrop One, a micro-UV/Vis spectrophotometer (Thermo Fisher), is used to measure the unbound con A in the supernatant. An absorbance based on the Protein A280 is measured with three technical replicates (**Fig 1A**).

## Preparation of testicular cells for binding assay

Mouse testicular cells are obtained by the following method (La Salle et al., 2009) with some modifications. Briefly, a C57BL/6J mouse at 20 days postpartum (dpp) is euthanized using $CO_2$ gas, and testes are detunicated. The seminiferous tubules are placed in a 20 mL of KRB medium (120 mM NaCl, 4.8 mM KCl, 25.2 mM $NaHCO_3$, 1.2 mM $KH_2PO_4$, 1.2 mM $MgSO_4 \cdot 7H_2O$, 1.3 mM $CaCl_2$, Pen/Strep (Gibco, 11548876), 11.1 mM D-glucose, essential amino acid (Gibco, 11130036), and nonessential amino acid (Gibco, 12084947), and containing collagenase type I (0.5 mg/mL, Wako, 035–17604), and incubated for 30 min at 32˚C with shaking at 250 rpm. The seminiferous tubules are washed twice with a KRB medium. The tubules are further digested in 20 mL of fresh KRB medium containing Trypsin (0.5 mg/mL Wako, 207–19982) and DNase I (1 µg/mL, Wako, 314–08071) for 15 min at 32˚C with shaking at 250 rpm to obtain a single-cell suspension. After a quick spin, cell pellets are washed twice with KRB medium containing 2% FBS and DNase I (1 µg/mL), and resuspended in 1× PBS (pH 6.8).

## Cell-binding assay

To obtain cultured cells, human lymphoma cell line Daudi (JCRB9071) and Large T-transformed human embryonic kidney cell line HEK293T (RIKEN BRC2202) are cultured in D-MEM medium (Wako, 044–29765) supplemented with 10% FBS, GlutaMAX (Gibco, 25050–061), Pen/Strep (Gibco, 15140–122) or RPMI-1640 medium (Wako, 189–02025) supplemented with 10% FBS, respectively, in a humidified air condition containing 5% $CO_2$ at 37˚C. 10 µL of each con A-conjugated magnetic beads are mixed with mouse testicular cells ($4 \times 10^5$ cells), Daudi ($5 \times 10^5$ cells), or HEK293T ($2 \times 10^5$ cells), respectively, and incubated for 30 min at RT. Cells bound to con A-conjugated beads are captured with a magnetic stand (FG-SSMAG2, FastGene), and numbers of con A beads-unbound cells in the supernatant are counted using a Scepter 2.0 automated cell counter (Merck) and 60 µL sensor (**Fig 1C**).

## CUT&Tag

CUT&Tag is performed according to the previous publication [3] and its updated modifications available online (www.protocols.io). Briefly, Tn5-adapter complex is prepared by annealing each of Mosaic end-adapter A (ME-A) and Mosaic end-adapter B (ME-B) oligonucleotides with Mosaic end-reverse oligonucleotides. To obtain Protein A-Tn5 fusion protein, the expression vector, 3XFlag-pA-Tn5-Fl (Addgene plasmid #124601), is transfected into *E. coli* strain BL21-pLysS Singles™ Competent Cells (Novagen, 70236) and cultured in 50 mL of LB medium containing ampicillin (100 µg/mL) at 180 rpm at 37˚C until OD600 = 0.9. Protein production is induced by adding IPTG (0.25 mM) and incubated at 180 rpm at 10˚C overnight, then at 180 rpm 23˚C for 4 hrs. The bacterial pellet is resuspended in 20 mL of HEGX buffer (20 mM HEPES-KOH (pH 7.2), 0.8 M NaCl, 1 mM EDTA, 10% glycerol, 0.2% Triton X-100) containing Complete Protease inhibitor cocktail (Roche), and lysed by sonication. After centrifugation

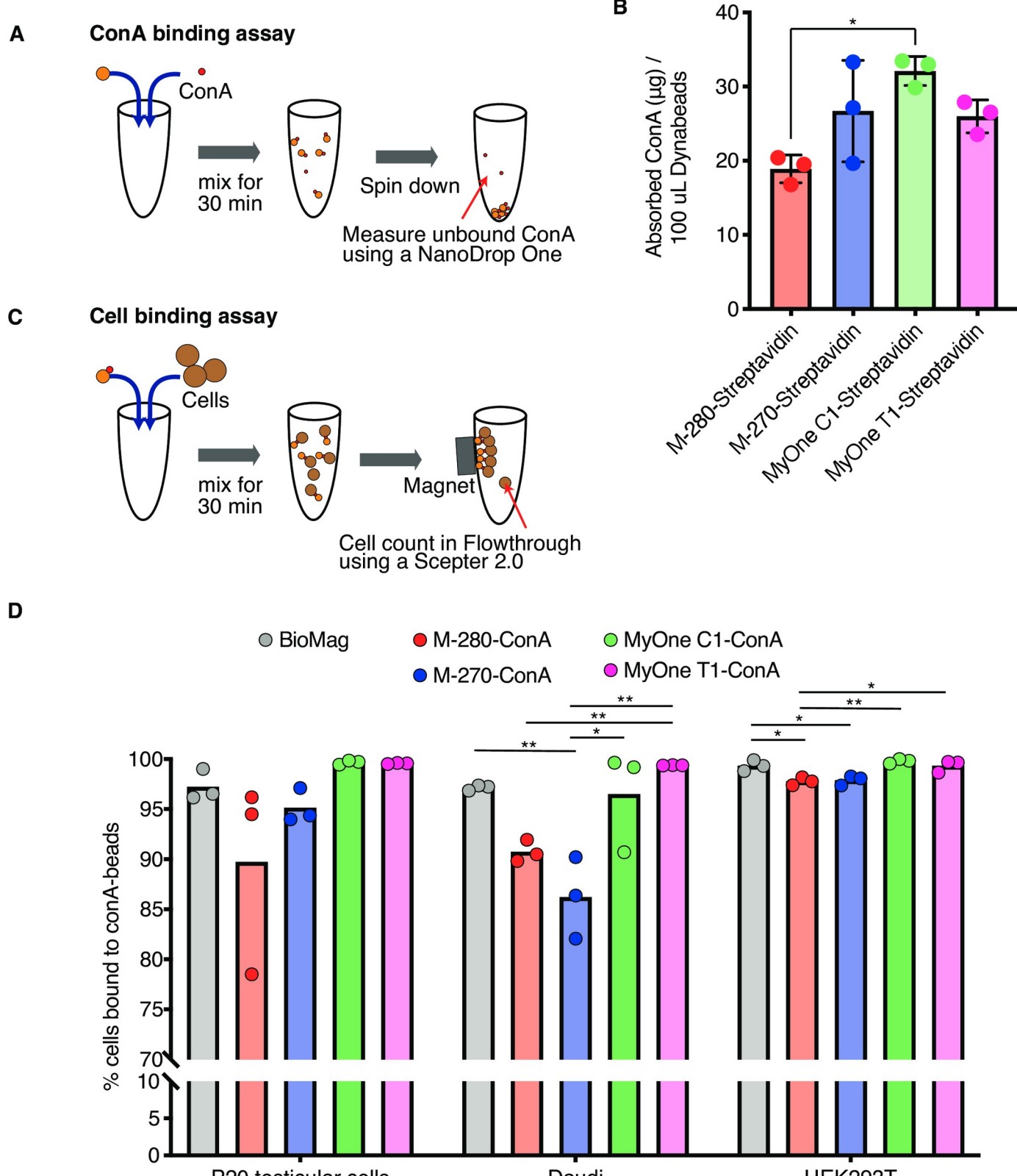

**Fig 1. Con A binding assay and cell binding assay. A,** Schematic view of con A binding assay. **B,** Con A-binding ability was compared between different magnetic beads. Columns indicate mean. Bars indicate SD. Dots indicate individual results of technical replicates. Statistical significance was assessed by one-way ANOVA followed by Turkey's multiple comparisons test with a single pooled variance. *: $P < 0.05$. N = 3. **C,** Schematic view of cell binding assay. **D,** Cell-binding ability of self-con A-conjugated Dynabeads magnetic beads and BioMag beads with three different kinds of cell types. Columns indicate mean. Dots indicate individual results of technical replicates. Statistical significance was assessed by one-way ANOVA followed by Turkey's multiple comparisons test with a single pooled variance. *: $P < 0.05$, **: $P < 0.01$. N = 3.

at 12,000 ×g at 4˚C for 30 min, 1/40 volume of 10% neutralized PEI is added, and incubated at 4˚C for 30 min. The collected supernatant is filtered through 0.45 μm mesh, and 7 mL of chitin slurry resin (NEB, S6651S) is added and incubated at 4˚C overnight. The fusion protein is eluted with total 17.5 mL of elution buffer (HEGX buffer containing 100 mM DTT), and dialyzed with 500 mL of Dialysis buffer (100 mM HEPES-KOH (pH 7.2), 100 mM NaCl, 0.2 mM EDTA, 2 mM 2-ME, 0.2% Triton X-100, 20% Glycerol). The fusion protein is further purified in IEX-A buffer using HiTrap SP1 and ACTA start (GE Healthcare Life Sciences). The Protein A-Tn5 fusion protein is mixed with pre-annealed ME-A and ME-B oligonucleotides. Ten μL (per sample) of either con A-conjugated Dynabeads MyOne T1 (hereafter MyOne T1) or Bio-Mag was added to 100 μL Binding buffer (20 mM HEPES pH 7.5, 10 mM KCl, 1 mM CaCl$_2$, 1 mM MnCl$_2$). After washing the beads with the Binding buffer twice, the beads are resuspended in a 10 μL of the Binding buffer. The beads slurry is then added to 0.2 mL of Wash buffer [20 mM HEPES pH 7.5, 150 mM NaCl, 0.5 mM spermidine, Complete Protease Inhibitor cocktail (Roche)]. This beads mixture is used to resuspend the harvested cells (~100,000 cells). The cell-containing tubes (two replicate samples are prepared for each assay) are rotated for 10 min at RT. After removing the buffer, the beads-bound cells are resuspended in 50 μL of ice-cold Antibody buffer (Wash buffer containing 0.05% digitonin, 2 mM EDTA, 0.1% BSA), and 0.5 μg of anti-H3K4me3 mouse monoclonal antibody (Abcam, ab12209) or the same amount of mouse IgG (BioLegend, 400101) is added to a tube. The tube is placed on an ELMI Intelli-mixer (PM-2m, ELMI) and mixed overnight at 4˚C (set with mode F8 at 30 rpm). After a quick spin, the tubes are placed on a magnet stand (12321D, Invitrogen), and the buffer is removed. The beads are then resuspended in 100 μL of Dig-wash buffer (Wash buffer containing 0.05% digitonin) containing the anti-mouse IgG (H&L) secondary antibody (1.1 μg, 610–4120, ROCKLAND), and placed on an ELMI Intelli-mixer (ELMI) and mixed for 60 min at RT. The beads are then repeatedly washed with a 0.2 mL of Dig-wash buffer 9 times. The washed beads are resuspended in 100 μL of the pA-Tn5 mix (final concentration 50 nM) containing Dig-300 buffer (20 mM HEPES pH 7.5, 300 mM NaCl, 0.5 mM spermidine, 0.01% digitonin, Complete Protease Inhibitor cocktail) (1:250 dilution of 12.5 μM pA-Tn5 stock), and gently mixed by vortexing. The tubes are placed on an ELMI Intelli-mixer (ELMI) and mixed for 1 hr at RT. After a quick spin, the tubes are placed on a magnet stand until the buffer turns clear. After removing the buffer, the beads are washed with 0.2 mL of Dig-300 buffer twice, and resuspended with 300 μL of Tagmentation buffer (Dig-300 buffer containing 10 mM MgCl$_2$), gently mixed, and incubated for 1 hr at 37˚C for adapter tagmentation reaction. To stop tagmentation and solubilize DNA fragments by de-crosslinking, 10 μL of 0.5 M EDTA, 3 μL of 10% SDS, and 2.5 μL of Proteinase K (20 mg/mL) are added to each sample and mixed by vortexing. The reaction tubes are then incubated at 37˚C overnight. After de-crosslinking, DNA is purified by Phenol-Chloroform-Isoamyl alcohol (25:24:1) extraction and Chloroform extraction followed by ethanol precipitation. Air-dried purified DNA is dissolved in 30 μL of 10 mM Tris-HCl pH 8.0 containing 1 mM EDTA and 25 μg/mL RNase A. The adapter sequence-tagmented DNA fragments are amplified by PCR using 21 μL DNA, 2 μL of Universal i5 primer (10 μM), 2 μL uniquely barcoded i7 primers (10 μM) [primer sequences are referred from [5]], and 25 μL NEBNext High-Fidelity 2x PCR Master mix (New England Bio-Labs, M0541S). The PCR cycle is as follows; 72˚C for 5 min, 98˚C for 30 sec, 98˚C for 10 sec, 63˚C for 10 sec, and the cycles are repeated 13 times, followed by 72˚C for 1 min. After cooling down, the PCR products are purified by using 55 μL of Ampure XP beads (1.1 volume of the PCR product) and cleaning with 80% ethanol. The cleaned DNA is then eluted with 25 μL of 10 mM Tris-HCl (pH 8.0). The size distribution of libraries is confirmed by capillary electrophoresis using an Agilent 2100 Bioanalyzer Instrument (Agilent Technologies) and High Sensitivity DNA Chips (5067–4626) (Agilent Technologies). The barcoded libraries are quantified

by KAPA Library Quantification Kit, Complete kit for ABI Prism, Illumina Platform (KK4835, 07960204001, Roche), and sample-unique primer sets are used for the amplification step. Paired-end sequencing of the barcoded libraries was performed using an Illumina HiSeq X sequencer (Illumina).

### Bioinformatics analysis

Barcode sequences of sequenced reads are eliminated using Trimmomatic [6], and the quality of reads is examined using FastQC [7]. Paired-end reads are aligned by Bowtie2 [8] to human genome GRCh37(hg19). The mapped data is converted to bam format using SAMtools [9], and narrow and broad peaks are detected using MACS [10]. Narrow peaks were detected by specifying the peak shift length 100 bp (default setting). For broad peak detection, the maximum shift length was set to 400 bp. Genome-wide distribution of peaks (bin size: 100,000), Heatmap, line plot and MA plot are made using DROMPA software [11]. Overlap of peaks is detected by using Intervene [12] and BEDTools [13]. To compare data between BioMag and MyOne T1 samples, merged peaks of each replicate sample are used. Bar graphs and MA plots are generated using GraphPad Prism 8 software.

## Result

### Con A binding assay

To quantify the binding capacities of Dynabeads to con A, we first measured the concentration of unbound con A in the supernatant after con A-Dynabeads coupling reaction (**Fig 1A**). The amount of absorbed (i.e., beads-bound) con A was calculated by subtracting the concentration of unbound con A from the initial concentration (2.3 μg/μL). Among the four tested Dynabeads, M-280 was less capable of binding to con A, while the other 3 types of Dynabeads (M-270, MyOne C1, and My One T1) exhibited comparable binding capacity in a range of 0.2–0.3 μg biotin-conjugated recombinant con A/μL beads solution (**Fig 1B**).

### Cell-binding assay

We next compared cell binding ability of the Dynabeads to three different cell types (Daudi, HEK293T, and mouse testicular cells). After removing con A-bound cells, the number of con A-unbound cells in the supernatant was counted (**Fig 1C**). We found that M-280 and M-270, which were bigger in diameter compared with those of MyOne C1 and T1 (2.8 μm vs. 1.05 μm), demonstrated relatively low capability to capture cells, especially for Daudi, although ~90% of cells were captured in all cell types (**Fig 1D**). Notably, the cell-binding ability of MyOne T1 and C1 beads appeared to be equivalent or even better than that of BioMag beads (1.05 μm) used in the original protocol (**Fig 1D**). Particularly, MyOne T1 exhibited the best performance in all cell types, indicating the usefulness of these small-sized Dynabeads for efficient and stable cell capture.

### CUT&Tag and data analyses

To further test the practicality of MyOne T1 in the CUT&Tag procedure, we next performed CUT&Tag in Daudi cells and compared the usability of BioMag and MyOne T1 beads. Previously, we have experienced the inconvenience of using BioMag beads during the CUT&Tag procedure in the following two points; one is the difficulty of uniform suspension of BioMag in the solution as it is easily aggregated after attaching cells to the beads (**Fig 2A**). Another is serious attachment to the wall of an Eppendorf tube. These issues could cause inefficient cell capture. The result clearly demonstrated that MyOne T1 exhibited more efficient and uniform

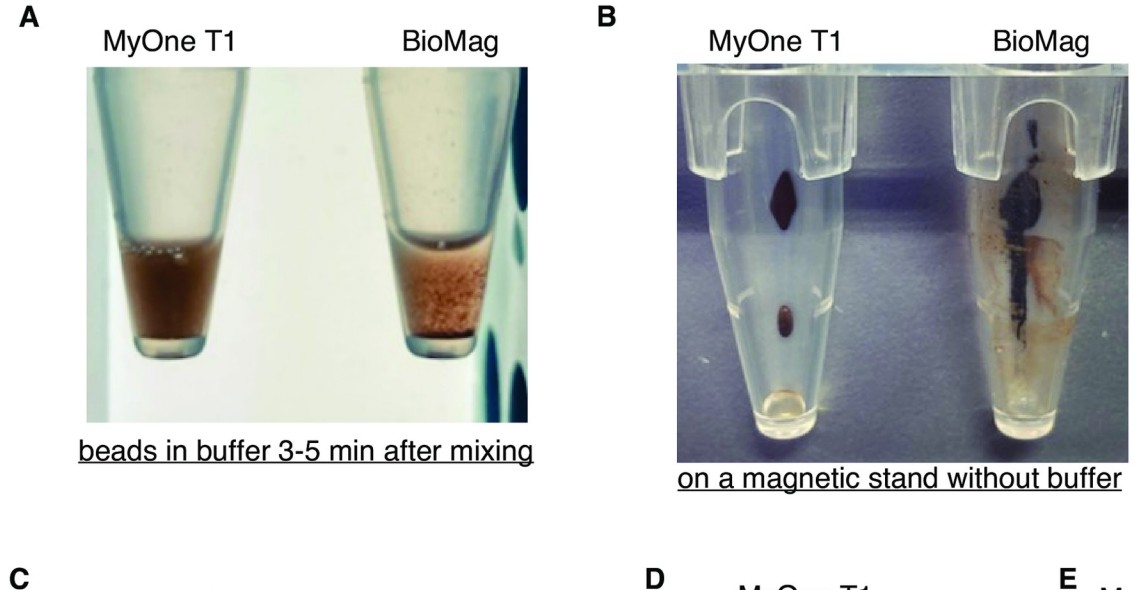

**Fig 2. CUT&Tag using BioMag and MyOneT1 beads. A,** Appearance of 1.5 mL tubes containing either BioMag or MyOne T1 on a magnetic stand just after removing the Binding buffer. **B,** Appearance of 1.5 mL tubes containing either BioMag or MyOne T1 in Antibody buffer 3–5 min after mixing. **C,** Appearance of 0.2 mL PCR tubes containing either BioMag or MyOne T1 that were bound with cells in Antibody buffer just after mixing (upper) and after ON incubation at 4°C (lower). **D,** Size distribution of libraries by capillary electrophoresis using Agilent 2100 Bioanalyzer Instrument. **E.** Electropherogram plots fluorescence intensity versus size (bp) for the sample indicated above. FU: Arbitrary fluorescence unit.

suspendability compared with BioMag, and there were few residual beads attached to the wall of a tube (**Fig 2B**). The beads-bound cells were then mixed with antibodies against H3K4me3 or control IgG and incubated overnight at 4˚C. After incubation, cell-bound MyOne T1 were uniformly dissolved in the buffer, while cell-bound BioMag was severely aggregated (**Fig 2C**). Subsequently, the size distribution and concentration of the library were analyzed by capillary electrophoresis. The original CUT&Tag using BioMag reported nucleosomal ladders when anti-H3K4me3 antibody is used [3]. In agreement with the original study, nucleosomal ladders were evidently observed in BioMag. Notably, MyOne T1 showed a reduced relative amount of DNA in higher ($> 1000$ bp) molecular weight, indicating the better performance of Tn-5 in the MyOne T1 samples (**Fig 2E**).

We next performed paired-end 150 bp-sequencing of the libraries that were made with either BioMag or MyOne T1 beads. Genome-wide distribution of the peaks was identified by comparing them with the IgG control samples. Overall, the peak distribution pattern was similar between BioMag and MyOne T1 beads at a chromosome-wide level (**Fig 3A**). In a 1–2 Mb window, H3K4me3 was observed as either narrow peaks (~100 bp, **Fig 3B**) or broad peaks (~400 bp, **Fig 3C**) depending on the genome loci. In both cases, the peak distribution and pattern were consistent between BioMag and MyOne T1 beads.

Further bioinformatic analyses demonstrated that 98.46% of narrow peaks and 99.26% of broad peaks from BioMag were overlapped with those from MyOne T1 beads (**Fig 4A**). In addition, peak distribution around genes was also similar between BioMag and MyOne T1 as the majority of H3K4me3 peaks were mainly distributed at upstream and genic regions rather than intergenic regions in both samples with a significant enrichment at transcription start sites (TSS) (**Fig 4B and 4E**). Furthermore, MyOne T1 showed notable numbers of MyOne T1-specific narrow and broad peaks (2723 and 1727 peaks, respectively) (**Fig 4A**), and these peaks contained relatively fewer upstream and more intergenic peaks as compared with the total MyOne T1 peaks (**Fig 4B**). In addition, the read density of T1 specific narrow peaks was evidently lower than that of T1/BM common peaks, and those of BM samples were even lower (**Fig 4C and 4D**).

Heatmaps and line plots of H3K4me3 peaks (both narrow and broad) from BioMag or MyOne-T1 showed a similar pattern (**Fig 4F and 4G**). MA plot analyses further indicated that 4.4–10.9% of the peaks were uniquely detected in MyOne T1, although their read counts exhibited relatively lower values (**Fig 4H**).

## Discussion

The present study showed that the use of Dynabeads, especially MyOne T1, can be better alternatives to BioMag used in an original CUT&Tag protocol in terms of easy handling and equivalent data quality [3,4]. Unpreferable aggregation and absorption to the wall of tubes that were frequently observed when BioMag was used can be minimized by using MyOne T1, potentially preventing the loss of cells and increasing the efficiency of the washing process. This advantage is probably because of the hydrophobic surface of MyOne T1. Interestingly, we also found that MyOne T1 or C1 magnetic beads, that were smaller in size compared with Dynabeads M-280 or M-270, showed slightly higher ability to capture three kinds of tested cells, suggesting that beads in a smaller size are more capable of binding cells due to the increased surface area of beads when the same volume of beads was used.

In addition, our subsequent CUT&Tag using the MyOne T1 followed by sequencing observed remarkable similarity in H3K4me3 peaks between MyOne T1 and BioMag (**Figs 3 and 4**). Importantly, 4.4–10.9% of the H3K4me3 peaks observed using MyOne T1 showed higher enrichment at detected peak regions than those of BioMag, while almost no peaks

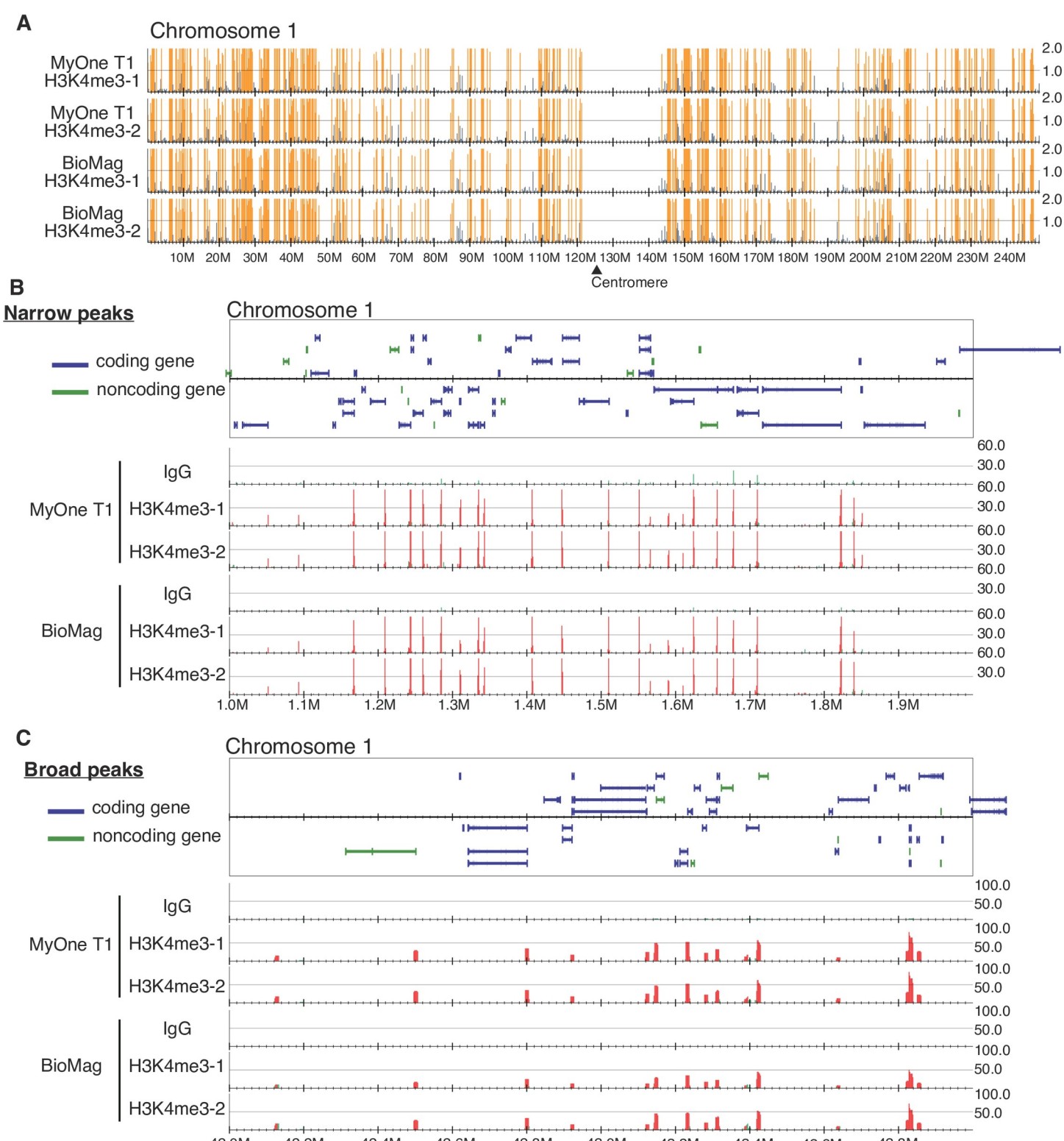

**Fig 3. Sequence results of libraries prepared from CUT&Tag. A,** Distribution of peaks of entire human chromosome 1. Yellow vertical lines indicate statistically significant peaks detected by DROMPA software. **B,** Distribution of Narrow peaks at gene level in the chromosome 1. Red vertical lines indicate statistically significant peaks detected by DROMPA software. **C,** Distribution of Broad peaks at gene level in the chromosome 1. Red peaks indicate statistically significant peaks detected by DROMPA software.

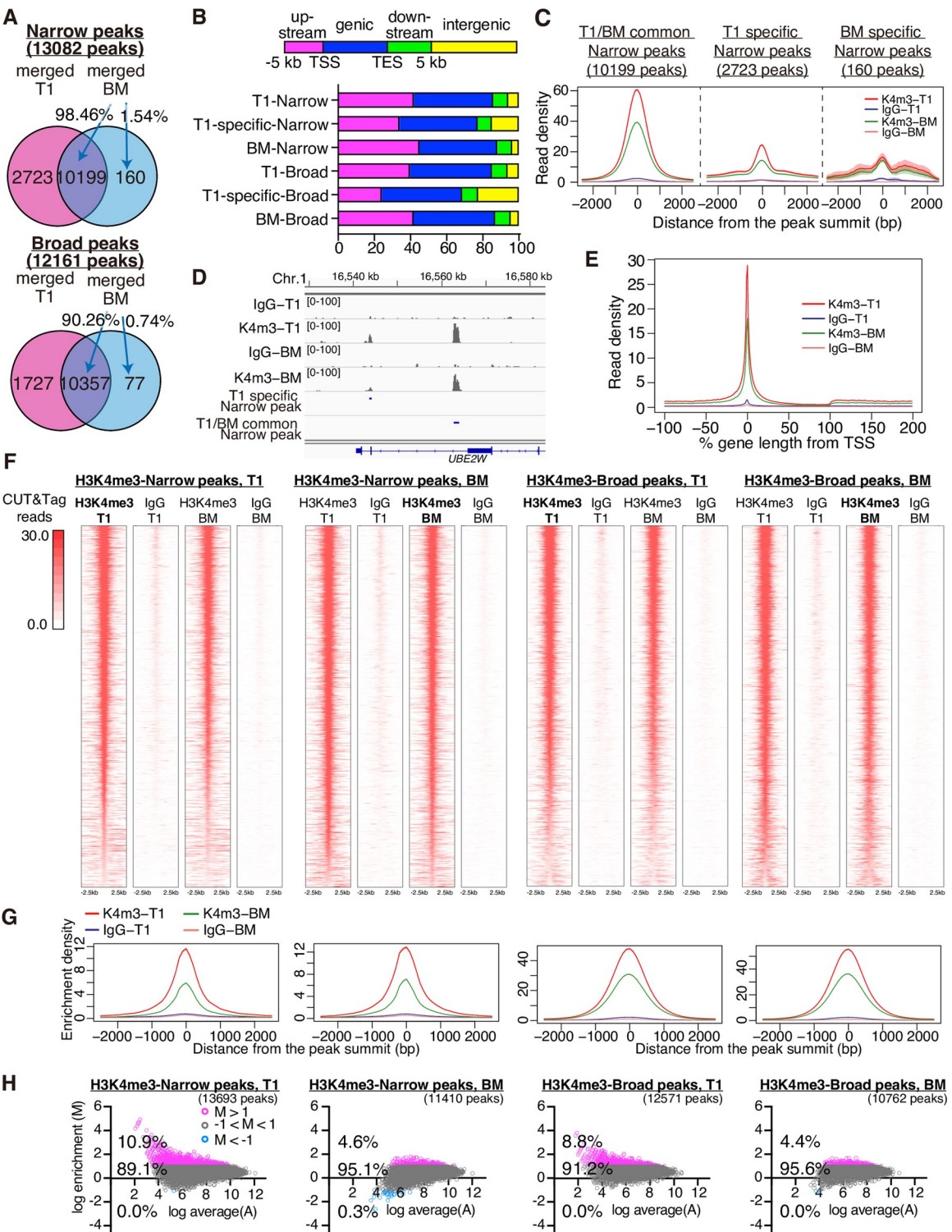

**Fig 4. Comparison of H3K4me3 peaks. A,** Venn diagram for overlap of merged Narrow and Broad peaks between MyOne T1 (T1) and BioMag (BM). Numbers indicate the number of peaks detected. **B,** Percentage of distribution of H3K4me3 reads in upstream (transcription start site, TSS –5 kb, magenta), genic (blue), downstream (transcription end site, TES +5 kb, green), and intergenic regions (yellow). **C,** Line plots show read density of T1/BM common or T1 specific narrow peaks from A. **D,** Distribution of mapped reads at T1 specific peaks and a T1/BM common peak on the chromosome 1 represents read densities for each peak. **E,** Enrichment profiles of CUT&Tag reads at TSS (0%). **F,** Heatmap analyses of CUT&Tag reads around H3K4me3 narrow or broad peaks detected using BM or T1 magnetic beads. **G,** Line plots below heatmaps represent CUT&Tag read density for each corresponding plot in F. **H,**

MA plots show ratio intensities to compare CUT&Tag read densities at T1 and BM peaks (for non-merged single replicate samples). Magenta circles represent log enrichment > 1, grey circles represent log enrichment log enrichment ±1, and blue ones represent low enrichment < -1.

showed lower enrichment (**Fig 4E**). Also, considering the increased number of H3K4me3 peaks in MyOne T1 samples (**Fig 4A**), using MyOne T1 may increase the sensitivity of the CUT&Tag. This may be because the improved suspendability by MyOne T1 enhanced the accessibility of IgG to the nucleus. On the other hand, MyOne T1 slightly increases the H3K4me3 peaks in intergenic regions (**Fig 4B**), implying the possibility of non-specific binding as well. Indeed, the read density of T1 specific narrow peaks was evidently lower than that of T1/BM common peaks, and those of BM samples were even lower (**Fig 4C and 4D**). Therefore, to verify the specificity of MyOne T1 beads, further examination is needed.

Nevertheless, we propose that the homemade con A-conjugated Dynabeads magnetic beads (i. e. MyOne T1) can be a "stress-free" alternative choice for CUT&RUN and CUT&Tag methods to avoid unwanted clumping and sticking to the sides of tubes. Preparation of the beads can be done fast and straightforward, thus most likely improving the quality of CUT&RUN or CUT&Tag results. It may need to test other types of magnetic beads to find the best ones for the cell types used in your experiments.

## Acknowledgments

We thank members of the Okada lab for their helpful discussion. We also thank Chie Kodama, (Veritas Corporation) for her technical advice.

## Author Contributions

**Conceptualization:** Yasuhiro Fujiwara, Yuji Tanno, Yoshinori Makino, Yuki Okada.

**Data curation:** Yasuhiro Fujiwara, Yuki Okada.

**Formal analysis:** Yasuhiro Fujiwara, Yuki Okada.

**Funding acquisition:** Yasuhiro Fujiwara, Yuki Okada.

**Investigation:** Yasuhiro Fujiwara, Hiroki Sugishita, Yusuke Kishi, Yoshinori Makino.

**Methodology:** Yasuhiro Fujiwara, Yuji Tanno, Yoshinori Makino, Yuki Okada.

**Resources:** Yasuhiro Fujiwara, Yuji Tanno, Yoshinori Makino, Yuki Okada.

**Supervision:** Yuki Okada.

**Validation:** Hiroki Sugishita, Yusuke Kishi.

**Visualization:** Yasuhiro Fujiwara, Yuji Tanno, Hiroki Sugishita, Yusuke Kishi, Yuki Okada.

**Writing – original draft:** Yasuhiro Fujiwara.

**Writing – review & editing:** Yasuhiro Fujiwara, Yuji Tanno, Hiroki Sugishita, Yusuke Kishi, Yuki Okada.

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
