## [Decision Letter · Decision Letter 0]

16 Jul 2021

PONE-D-21-13514

Preparation of "stress-free" concanavalin A-conjugated Dynabeads® magnetic beads for CUT&Tag

PLOS ONE

Dear Dr. Fujiwara,

Thank you for submitting your manuscript to PLOS ONE. After careful consideration, we feel that it has merit but does not fully meet PLOS ONE’s publication criteria as it currently stands. Therefore, we invite you to submit a revised version of the manuscript that addresses the points raised during the review process.

We look forward to receiving your revised manuscript.

Kind regards,

Nazmul Haque

Academic Editor

PLOS ONE

Journal Requirements:

"This work was supported in part by JSPS KAKENHI grant number 17K15392 and

20H00446 (to Y.F.) and JST ERATO (JPMJER1901 to Y.O.)."

We note that one or more of the authors are employed by a commercial company: "VERITAS Corporation"

Reviewers' comments:

Reviewer's Responses to Questions

**Comments to the Author**

1. Is the manuscript technically sound, and do the data support the conclusions?

Reviewer #1: Yes

Reviewer #2: No

2. Has the statistical analysis been performed appropriately and rigorously? 

Reviewer #1: N/A

Reviewer #2: Yes

3. Have the authors made all data underlying the findings in their manuscript fully available?

Reviewer #1: Yes

Reviewer #2: Yes

4. Is the manuscript presented in an intelligible fashion and written in standard English?

Reviewer #1: Yes

Reviewer #2: Yes

5. Review Comments to the Author

Reviewer #1: The authors test different concanavalin A coated magnetic beads for CUT&RUN and CUT&Tag and find that the commercial beads recommended for these protocols are not ideal in that they may cause clumping and sticking to the sides of tubes. The authors test various versions of Dynabeads and find that the hydrophobic product solves the problem. Derivatizing Dynabeads with ConA is fast and simple, and given the improved results, it should be easily adopted by labs using these methods, and is likely to become popular. The work is clearly presented and conservatively interpreted.

Reviewer #2: The manuscript “Preparation of "stress-free" concanavalin A-conjugated Dynabeads® magnetic beads for CUT&Tag” comparing different formulations of magnetic beads for cell handling in CUT&Tag chromatin profiling. The basic CUT&Tag protocol for chromatin profiling is increasingly popular in many laboratories. In standard protocols commercial magnetic beads coupled to concanavalin A are used to capture cells for processing. The authors develop a procedure for making bead preparations from commercial strptavidin-coated magnetic beads and commercial biotinylated concanavalin A. The authors describe beads that are pretty suitable for CUT&Tag, with better suspension characteristics. However, they claim that these bead preparations improve the data quality of CUT&Tag chromatin profiling. This claim needs additional analysis for support.

Line 44-45 states that “...the 45 MyOne-T1 beads notably improved the sensitivity in CUT&Tag assay for H3K4me3.” However, the data presented in Figure 4 shows that MyOne T1 beads have substantially increased background signal, seemingly twice as much as the BioMag, with only best a ~10% gain in signal. This is most obvious in Figure 4C and 4D. Quantitation of signal and background should be shown to support the conclusion that data quality is improved.

The meaning of “stress-free” in the title is not clear and should be removed. At first this seemed to refer to a more gentle reagent for sensitive cells, but this term is only mentioned in the last sentence “...we propose that the home-made con A-conjugated Dynabeads magnetic beads (i. e. MyOne T1) can be a "stress-free" alternative choice for CUT&RUN and CUT&Tag methods.” Is this stress-free for the experimenter? If not, it should be clarified.

6. PLOS authors have the option to publish the peer review history of their article (what does this mean?). If published, this will include your full peer review and any attached files.

Reviewer #1: **Yes: **Steven Henikoff

Reviewer #2: No

---

## [Author Response · Author response to Decision Letter 0]

10 Oct 2021

Response to reviewers’ comments

August 30, 2021

Dear Reviewers,

We thank the reviewers for their careful reading of the manuscript and their thoughtful comments and suggestions. Also, we greatly appreciate the editor's decision to give us a chance to submit the revised manuscript. Our point-to-point responses to editorial and reviewers' comments are shown below. We believe our revised manuscript would suffice the reviewer's concerns. In the revised manuscript, the sentences added or changed are shown as marked-up of Track Change function of Word in the main text titled "Revised Manuscript with Track Changes". It would be greatly appreciated if the editorial office and reviewers could consider this revision.

Reviewer #1: The authors test different concanavalin A coated magnetic beads for CUT&RUN and CUT&Tag and find that the commercial beads recommended for these protocols are not ideal in that they may cause clumping and sticking to the sides of tubes. The authors test various versions of Dynabeads and find that the hydrophobic product solves the problem. Derivatizing Dynabeads with ConA is fast and simple, and given the improved results, it should be easily adopted by labs using these methods, and is likely to become popular. The work is clearly presented and conservatively interpreted.

Response: We appreciate the positive evaluation, which requires no specific response.

Reviewer #2: The manuscript “Preparation of "stress-free" concanavalin A-conjugated Dynabeads® magnetic beads for CUT&Tag” comparing different formulations of magnetic beads for cell handling in CUT&Tag chromatin profiling. The basic CUT&Tag protocol for chromatin profiling is increasingly popular in many laboratories. In standard protocols commercial magnetic beads coupled to concanavalin A are used to capture cells for processing. The authors develop a procedure for making bead preparations from commercial strptavidin-coated magnetic beads and commercial biotinylated concanavalin A. The authors describe beads that are pretty suitable for CUT&Tag, with better suspension characteristics. However, they claim that these bead preparations improve the data quality of CUT&Tag chromatin profiling. This claim needs additional analysis for support.

Response: We appreciate the reviewer for the positive evaluation and valuable comments. As suggested, we re-analyzed the data and added some new data, which improved the manuscript. 

Comment #1: Line 44-45 states that “...the 45 MyOne-T1 beads notably improved the sensitivity in CUT&Tag assay for H3K4me3.” However, the data presented in Figure 4 shows that MyOne T1 beads have substantially increased background signal, seemingly twice as much as the BioMag, with only best a ~10% gain in signal. This is most obvious in Figure 4C and 4D. Quantitation of signal and background should be shown to support the conclusion that data quality is improved.

Response: We appreciate for bringing the background issue to our attention. Some of the figure panels, especially old Fig4C, can mislead the readers. Using our con-A beads, more antibody was presumably introduced into cells efficiently, probably increasing the antibody's non-specific binding. Therefore, as you suggested, our data clearly showed higher background in T1 samples than in BioMag samples, especially at those peaks with high enrichment. However, those background signals were considerably low compared with H3K4me3 reads, and the S/N ratio is remarkably high. Because the y axis of the line plot in the old Fig4C was shown in the log scale, we changed it to linear (panel E in the new figure 4). We further re-analyzed the T1 and BioMag peaks and found that at T1-specific peaks, both T1 and BM samples showed read enrichment at a certain level. However, the S/N ratio for the BM sample was lower and not enough to be detected as a peak (new Fig4C, D). Further experiments are required to answer whether these peaks have biological meanings or background noises remain elusive.

Comment #2: The meaning of “stress-free” in the title is not clear and should be removed. At first this seemed to refer to a more gentle reagent for sensitive cells, but this term is only mentioned in the last sentence “...we propose that the home-made con A-conjugated Dynabeads magnetic beads (i. e. MyOne T1) can be a "stress-free" alternative choice for CUT&RUN and CUT&Tag methods.” Is this stress-free for the experimenter? If not, it should be clarified.

Response: We agree with the reviewer's suggestions and removed the "stress-free" from the title. We instead added 'optimized' to emphasize the advantages of the beads. In addition, the last paragraph of the discussion part was changed from “...we propose that the home-made con A-conjugated Dynabeads magnetic beads (i. e. MyOne T1) can be an alternative "stress-free" choice for CUT&RUN and CUT&Tag methods.” to “...we propose that the home-made con A-conjugated Dynabeads magnetic beads (i. e. MyOne T1) can be an alternative "stress-free" choice for CUT&RUN and CUT&Tag methods to avoid unwanted clumping and sticking to the sides of tubes. Furthermore, preparation of the beads can be done fast and simple, thus most likely improve the quality of CUT&Tag or CUT&RUN results.”. This summarizes why and how this method improves CUT&Tag method with minimal addition of steps.

---

## [Decision Letter · Decision Letter 1]

28 Oct 2021

Preparation of optimized concanavalin A-conjugated Dynabeads® magnetic beads for CUT&Tag

PONE-D-21-13514R1

Dear Dr. Fujiwara,

We’re pleased to inform you that your manuscript has been judged scientifically suitable for publication and will be formally accepted for publication once it meets all outstanding technical requirements.

Kind regards,

Nazmul Haque

Academic Editor

PLOS ONE

Additional Editor Comments (optional):

Reviewers' comments:

Reviewer's Responses to Questions

**Comments to the Author**

1. If the authors have adequately addressed your comments raised in a previous round of review and you feel that this manuscript is now acceptable for publication, you may indicate that here to bypass the “Comments to the Author” section, enter your conflict of interest statement in the “Confidential to Editor” section, and submit your "Accept" recommendation.

Reviewer #1: All comments have been addressed

Reviewer #2: All comments have been addressed

2. Is the manuscript technically sound, and do the data support the conclusions?

Reviewer #1: Yes

Reviewer #2: Yes

3. Has the statistical analysis been performed appropriately and rigorously? 

Reviewer #1: N/A

Reviewer #2: Yes

4. Have the authors made all data underlying the findings in their manuscript fully available?

Reviewer #1: Yes

Reviewer #2: Yes

5. Is the manuscript presented in an intelligible fashion and written in standard English?

Reviewer #1: Yes

Reviewer #2: Yes

6. Review Comments to the Author

Reviewer #1: The authors' responses are satisfactory. This contribution represents a useful modification of a method that many people are beginning to use. The manuscript is ready for publication in my opinion.

Reviewer #2: (No Response)

7. PLOS authors have the option to publish the peer review history of their article (what does this mean?). If published, this will include your full peer review and any attached files.

Reviewer #1: **Yes: **Steven Henikoff

Reviewer #2: No

---

## [Editor Report · Acceptance letter]

2 Nov 2021

PONE-D-21-13514R1 

Preparation of optimized concanavalin A-conjugated Dynabeads® magnetic beads for CUT&Tag 

Dear Dr. Fujiwara:

I'm pleased to inform you that your manuscript has been deemed suitable for publication in PLOS ONE. Congratulations! Your manuscript is now with our production department. 

Kind regards, 

on behalf of

Dr. Nazmul Haque 

Academic Editor

PLOS ONE